# Redox-coupled proton pumping drives carbon concentration in the photosynthetic complex I

Jan M. Schuller [1*], Patricia Saura [2,3,7], Jacqueline Thiemann [4,7], Sandra K. Schuller [1], Ana P. Gamiz-Hernandez[2,3], Genji Kurisu [5,6], Marc M. Nowaczyk [4*] & Ville R.I. Kaila [2,3*]

Photosynthetic organisms capture light energy to drive their energy metabolism, and employ the chemical reducing power to convert carbon dioxide ($CO_2$) into organic molecules. Photorespiration, however, significantly reduces the photosynthetic yields. To survive under low $CO_2$ concentrations, cyanobacteria evolved unique carbon-concentration mechanisms that enhance the efficiency of photosynthetic $CO_2$ fixation, for which the molecular principles have remained unknown. We show here how modular adaptations enabled the cyanobacterial photosynthetic complex I to concentrate $CO_2$ using a redox-driven proton-pumping machinery. Our cryo-electron microscopy structure at 3.2 Å resolution shows a catalytic carbonic anhydrase module that harbours a $Zn^{2+}$ active site, with connectivity to proton-pumping subunits that are activated by electron transfer from photosystem I. Our findings illustrate molecular principles in the photosynthetic complex I machinery that enabled cyanobacteria to survive in drastically changing $CO_2$ conditions.

[1] Department of Structural Cell Biology, Max Planck Institute of Biochemistry, 82152 Martinsried, Germany. [2] Department of Biochemistry and Biophysics, Stockholm University, SE-106 91 Stockholm, Sweden. [3] Center of Integrated Protein Science at the Department of Chemistry, Technical University of Munich, Garching, Germany. [4] Plant Biochemistry, Faculty of Biology and Biotechnology, Ruhr University Bochum, 44780 Bochum, Germany. [5] Institute for Protein Research, Osaka University, Suita, Osaka 565-0871, Japan. [6] Department of Macromolecular Science, Graduate School of Science, Osaka University, Toyonaka 560-0043, Japan. [7]These authors contributed equally: Patricia Saura, Jacqueline Thiemann *email: janschu@biochem.mpg.de; marc.m.nowaczyk@rub.de; ville.kaila@dbb.su.se

C yanobacteria evolved around 2.7 billion years ago with the ability to oxidise water into dioxygen ($O_2$) using the energy captured from sunlight[1]. The released $O_2$ powers respiratory chains in aerobic life, and the electrons extracted from the water oxidation are used for synthesis of organic molecules[2]. The light-driven water splitting catalysed by photosystem II (PSII), reduces plastoquinone (PQ) and establishes an electrochemical proton gradient across the thylakoid membrane that subsequently drives synthesis of adenosine triphosphate (ATP)[3,4]. During linear electron flow (LEF), the electrons are transferred to photosystem I (PSI), providing the reducing power for photosynthetic $CO_2$ fixation that consumes nicotinamide adenine dinucleotide phosphate (NADPH) and drives the synthesis of complex organic compounds from inorganic carbon ($C_i$)[2]. Photosynthetic organisms also employ cyclic electron flow (CEF) around PSI to increase the ATP/NADPH ratio that powers the $CO_2$ fixation.

$CO_2$ concentrations drastically changed during the last 2.7 billion years, from >0.5% (5000 ppm) to today's levels around 0.041% (410 ppm)[5]. To survive, cyanobacteria evolved carbon-concentrating mechanisms (CCMs) that enhance the efficiency of the photosynthetic $CO_2$ fixation process[6,7]. In many organisms, hydration of the gaseous $CO_2$ to the soluble bicarbonate ($HCO_3^-$) is catalysed by carbonic anhydrase (CA), one of the fastest known enzymes with a $k_{cat} \sim 10^6$ s$^{-1}$ [8]. However, cyanobacteria lack genes for canonical cytoplasmic CAs, and if artificially expressed in the cytoplasm, the bacteria do not survive in low $CO_2$ concentrations < 20 ppm[7]. Cyanobacteria express instead the inducible NDH-1MS (NDH-1$_3$) and constitutive NDH-1MS' (NDH-1$_4$) photosynthetic complex I isoforms that convert $CO_2$ into $HCO_3^-$ by kinetically shifting the reaction equilibrium towards bicarbonate, against high cytoplasmic $HCO_3^-$ concentrations[7,9]. The $HCO_3^-$ subsequently diffuses into the carboxysome micro-compartments, where it is converted by a carboxysomal CA to $CO_2$[7], which further carboxylates ribulose-1,5-bisphosphate (RuBP) into carbohydrates by the action of RuBisCO (Ribulose-1,5-bisphosphate carboxylase/oxygenase)[10]. This CCM of the photosynthetic complex I, prevents $CO_2$ to diffuse out of the cell by concentrating the $C_i$ for RuBisCO, providing a basis for the efficient carbon fixation that is hampered during photorespiration[7].

To determine the molecular architecture of NDH-1MS (NDH-1$_3$), we isolate the enzyme from the cyanobacterium *Thermosynechococcus elongatus*, solved its molecular structure at 3.2 Å resolution using cryo-EM (Fig. 1, Supplementary Fig. 1, Supplementary Table 1, and Supplementary Movie 1), and probe its molecular mechanism by classical and quantum mechanical simulations.

## Results

**Architecture of the photosynthetic complex I.** The 0.5 MDa complex has an overall U-shape with 19 isolated subunits (Supplementary Fig. 2 and Supplementary Table 2). The structure of NdhV could not be resolved. The core structure is highly conserved across the complex I superfamily[11–14], but in contrast to the respiratory enzyme, the photosynthetic complex lacks the N-module that accepts electrons from nicotinamide adenine dinucleotide (NADH) (Fig. 1a–d). Instead, the electrons directly enter a chain of three iron-sulphur (FeS) centres in the ferredoxin (Fd)-binding domain (Fig. 1d), similar to the recently characterised NDH-1L type photosynthetic complex I and membrane-bound hydrogenases[15–18]. The PQ-binding site is located ca. 20 Å above the membrane plane (Fig. 1d and Supplementary Fig. 3a–c, f), and the modelled PQ tail extends into the lipid membrane in the vicinity of the NdhL subunit, with experimentally resolved lipids stabilising the PQ entry gate (Supplementary Fig. 3b, f).

The modular membrane domain extends up to 200 Å away from the PQ reduction site, and it comprises the antiporter-like subunits NdhA, NdhB, NdhD3, and NdhF3, as well as the smaller transmembrane (TM) subunits NdhC/E/G/L. These subunits contain a chain of buried charged residues that establish central elements of the proton-pumping machinery[11–17,19,20], with the isoform-specific NdhD3 and NdhF3 subunits at the terminal end of the enzyme (Fig. 1d, see below).

**Structure and function of the $CO_2$ concentrating module.** The $CO_2$-concentrating CupA/S module (CO$_2$ uptake, Cup/CO$_2$ hydration protein, ChpY) is located on the cytoplasmic side of the membrane, extending up to 60 Å above the membrane plane (Figs. 1d and 2a and Supplementary Fig. 3d, e). The catalytic site responsible for the $CO_2$ chemistry in the 50 kDa CupA protein electrostatically binds NdhF3 (Fig. 1 and Supplementary Fig. 4a). CupA has a fold, comprising α-helices, which drastically differs from known carbonic anhydrases (CAs) in which the catalytic zinc ($Zn^{2+}$) ion binds to histidine (αCA family) or histidine/cysteine residues (βCA family) at the interface of β-sheets (Fig. 2a)[8]. CupA is capped by the 16 kDa α-helical CupS subunit that binds two helical lobes of the former, and could help in stabilising the CupA-NdhF3 interaction (Supplementary Fig. 4a). CupS undergoes a conformational change from its solution structure upon binding to CupA in which an α-helical lid (α2, α3) closes upon a β-sheet (Supplementary Fig. 4b)[21]. The active site of CupA shows a strong electron density that we assign to a $Zn^{2+}$ ion (Fig. 2a), as also supported by inductively-coupled plasma optical emission spectrometry (ICP-OES), which indicates a protein-to-zinc ratio of ~0.7 in the NDH-1MS sample (Supplementary Fig. 2e). The $Zn^{2+}$ ion is coordinated by His130, a putative water/hydroxide ligand or carbonate, and Arg135 (Fig. 2a). Our mass spectrometric (MS) measurements reveal no indication for citrullination of the arginine (Supplementary Fig. 2d, e), and our electrostatic calculations suggest that the residue is neutral with p$K_a$ << 7, due to its proximity to the $Zn^{2+}$ ion (Fig. 2a and Supplementary Fig. 5). Such $Zn^{2+}$-Arg coordination is rare, but similar structures are also found in a human genetic variant of carbonic anhydrase I (αCA type I)[22].

The $Zn^{2+}$-site is located at the interface of two 12 Å helices, and CupA forms further contacts with NdhF3 and several lipid molecules (Fig. 3a, b and Supplementary Fig. 3d, e). The putative $CO_2$ binding pocket is completed by residues from NdhF3: the $Zn^{2+}$-$H_2O$/$OH^-$ is stabilised by Arg37 that forms an ion-pair with Glu114 and Tyr41 of NdhF3. A non-polar tunnel starting from NdhF3 that leads to the $Zn^{2+}$-site via this Arg37/Glu114/ Tyr41 network could function as a $CO_2$ conduction channel from the luminal side across the membrane (Fig. 3c). Remarkably, NdhF3 binds a chlorophyll *a*/β-carotene motif that connects via His469 of NdhF3 in a TM-helix to the CupA site (Fig. 1c and Supplementary Fig. 3d). Despite lacking experimental evidence of its functionality, we speculate that the motif could be involved in light-triggered regulation of the CA activity and/or to provide structural stability. A similar Chl *a*/β−carotene motif with unknown function is also found in cytochrome $b_6f$[23].

**Mechanism of the $CO_2$ hydration reaction.** To probe the catalytic properties of the CupA module, we performed quantum chemical density functional theory (DFT) calculations on the $CO_2$ hydration reaction and compared this to the reaction energetics in αCA and βCA (Fig. 2b and Supplementary Fig. 5, see Methods). The $CO_2$ hydration is initiated by proton transfer (pT) from the Zn-bound water molecule followed by a nucleophilic attack of the hydroxide on $CO_2$. In αCA (βCA), the rate-limiting reaction barrier of 11 (12) kcal mol$^{-1}$ is connected with pT to His64

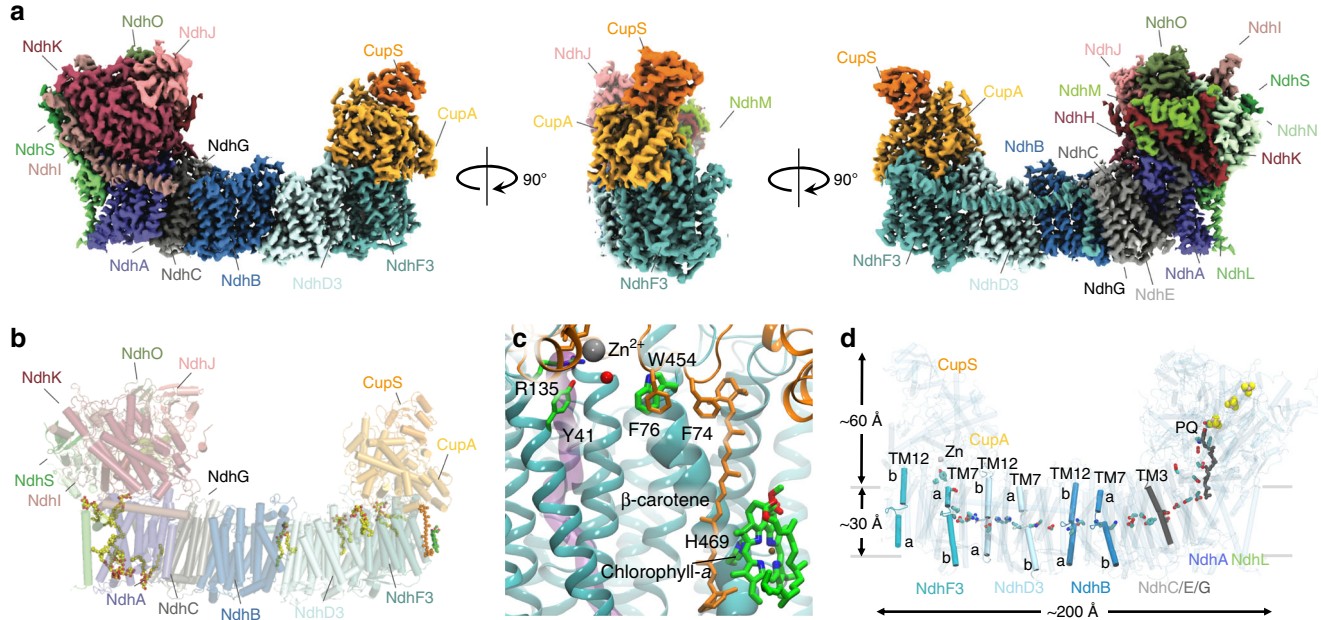

**Fig. 1 Structure of the carbon-concentrating photosynthetic complex I. a** The electron density map of NDH-1MS (NDH-1$_3$) shown from a back view (left), side view (middle), and front view (right). **b** Experimentally refined lipid molecules and cofactors in the hydrophobic domain of NDH-1MS. **c** The structure of the Chl *a/β*-carotene motif of NdhF3 that connects to the active site of CupA. See Supplementary Fig. 3 for example densities. **d** Chain of charged elements transmitting the redox-signal into proton pumping and CO$_2$ uptake. The PQ was computationally modelled into the experimental structure.

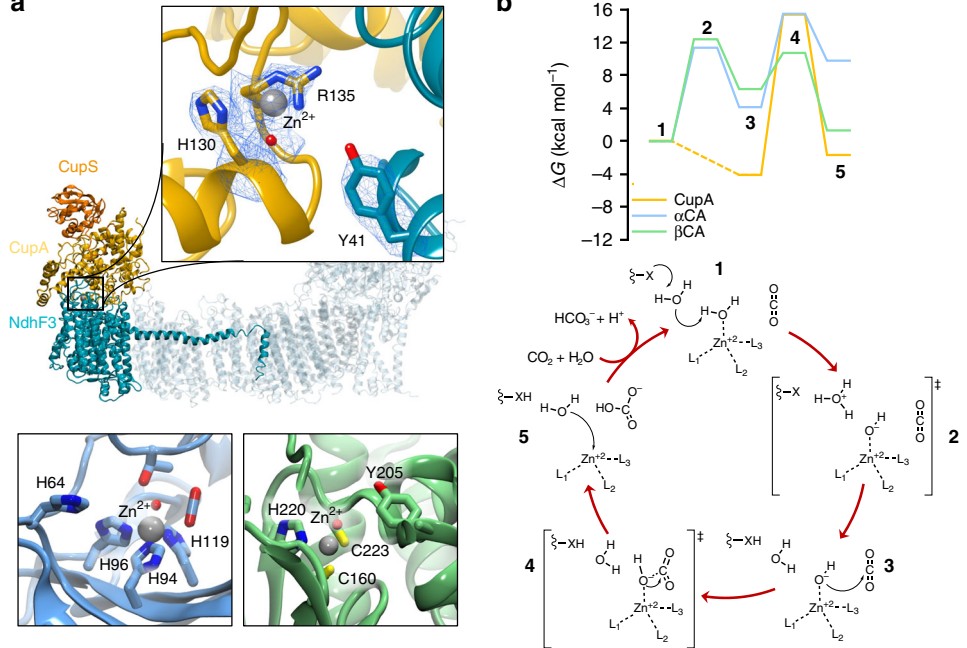

**Fig. 2 Structural comparison and mechanism of carbon concentration in CupA and canonical carbonic anhydrases. a** The active site structure of the CO$_2$ concentrating CupA subunit, showing the density of Zn-coordinating residues (5.5 sigma value, contour level 2, top), α-carbonic anhydrase (αCA, PDB ID: 5YUI, bottom left), and β-carbonic anhydrase (βCA, PDB ID: 1EKJ, bottom right). The Zn-bound density has been modelled as a water ligand, although the character of the ligand cannot be unambiguously assigned based on the map. **b** Reaction mechanism and free energy profiles for the CO$_2$ hydration process based on quantum chemical DFT models in CupA, αCA, and βCA with Tyr41, His64, and Tyr205 as proton acceptors, respectively. Free energies are reported at the B3LYP-D3/def2-TZVP/def2-SVP/$\varepsilon = 4$ theory level (see Methods). Ligands L$_{1/2/3}$ = His130/Arg135/H$_2$O, L$_4$ = H$_2$O/OH$^-$, X = Tyr41 in CupA; L$_{1/2/3}$ = His94/96/119, L$_4$ = H$_2$O/OH$^-$, X = His64 in αCA; L$_{1/2/3}$ = Cys160/223/His220, L$_4$ = H$_2$O/OH$^-$, X = Tyr205 in βCA. The CupA reaction takes place via two water molecules, modelled based on MD simulations (Supplementary Fig. 5j). See Supplementary Fig. 5 for further details.

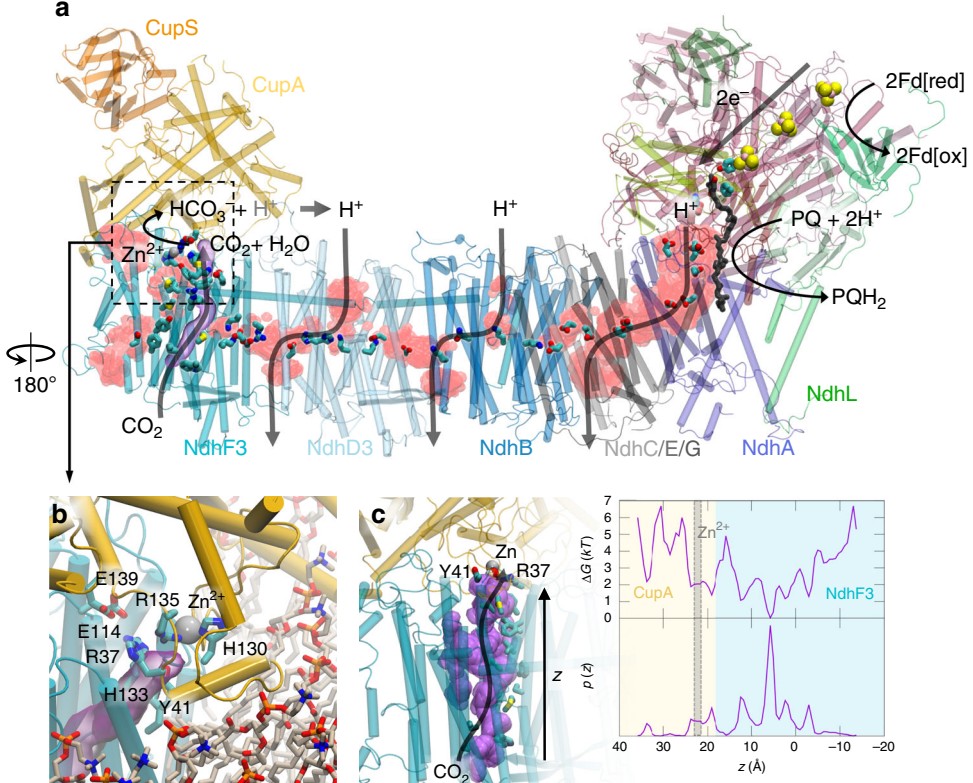

**Fig. 3 Proton pathways and putative CO$_2$ channel in the photosynthetic complex I. a** Location of proton pathways formed by water molecules (in red) during 250 ns atomistic molecular dynamics simulations. The figure shows the overall water density during MD. **b** Closeup of the active site of CupA at the NdhF3 interface (in cyan) and the lipid membrane from MD simulations. The putative CO$_2$ channel (in purple), visualised based on CAVER analysis (see Methods), is surrounded by hydrophobic and bulky residues, and connects to the Zn$^{2+}$-binding site by conserved Tyr41, Arg37, and Glu114 of the NdhF3 subunit (see Supplementary Fig. 6g for details). **c** Diffusion pathway of CO$_2$ molecules from the luminal side to CupA during MD simulations. Left: Dynamics of CO$_2$ in its putative gas channel in NdhF3. The average of the CO$_2$ position sampled during dynamics is represented as a purple surface. Right: Probability distribution and resulting potential of mean force (pmf in units of $kT$), $G(z) = -RT \log(p(z))$, of CO$_2$ inside the channel along the membrane axis ($z$-axis) obtained from 200 ns MD simulation. Owing to sampling gaps, the pmf is subjected to large errors.

(Tyr205), which compares well with the experimentally-observed barriers of ca. 10–12 kcal mol$^{-1}$ (Fig. 2b)[8]. In CupA, proton transfer from Zn-bound water to Tyr41 is slightly exergonic in our DFT models, whereas the nucleophilic attack of the Zn-bound OH$^-$ to the CO$_2$ has a barrier of ca. 15 kcal mol$^{-1}$, predicting that catalysis takes place in the millisecond timescale. The involvement of Tyr41, similarly to Tyr205 in βCA[24], is also supported by QM/MM models (Supplementary Fig. 5e), as well as by MD simulations and p$K_a$ calculations (Supplementary Fig. 5f). These findings suggest that our resolved structure is catalytically efficient, with kinetic barriers similar to the canonical CAs, despite its unique molecular architecture. CupA becomes well-hydrated during the MD simulations, establishing protonic connectivity with the NdhD3 subunit (Fig. 3a and Supplementary Figs. 5d and 6c), and with global dynamics inferred from the local resolution of the cryo-EM map, closely resembling the motion extracted from the simulations (Supplementary Fig. 7). The proton channels are established across the membrane around charged residues in the broken helices TM7 and TM12 of the antiporter-like subunits NdhB, NdhD3[19,20], and also in NdhA/C/E/G (Figs. 1d and 3, and Supplementary Fig. 6a–d, f) at locations where conformational changes were recently observed in the mammalian enzyme (Supplementary Fig. 6e)[12]. The structural architecture thus supports that the long-range protonation signal could be triggered by dissociation of conserved ion-pairs in the antiporter-like subunits that leads to lateral proton transfer in the proton channels by coupled conformational and hydration changes (Fig. 4)[16,19,20].

## Discussion

The NdhF3 comprises conserved ion-pairs at the interface to the NdhD3 subunit, similar as in the other antiporter-like subunits. However, charged elements in the lateral proton channel in NdhF3 are replaced by non-polar residues that could be employed to channel CO$_2$ into CupA. Indeed, we observe that CO$_2$ effectively diffuses along this channel to the active site of the CupA subunit in MD simulations (Fig. 3c and Supplementary Fig. 6g). Although the solubility of CO$_2$ is similar in lipid membranes and water[25], the gas tunnel could channel CO$_2$ into the active site of the CupA subunit, and prevent its escape under low CO$_2$ and high bicarbonate concentrations, arising from the substrate channelling to RuBisCO and bicarbonate pumps in the plasma membrane[7,26]. The thermodynamically unfavourable CO$_2$ hydration reaction is thus kinetically gated in the bicarbonate direction when the protons released in the CO$_2$ hydration process are pumped across the membrane by the antiporter-like subunits (Fig. 4). The driving force for this is achieved by the ca. 1070 mV redox span transduced from the PQ oxido-reduction chemistry ($E_m$(Fd) ≈ −420 mV[27]; $E_m$(PQ) ≈ +117 mV[28]), that is transmitted to the terminal edge of the membrane domain (Fig. 4)[19,20]. The carbon concentration thus couples to generation of proton motive force (pmf) across the membrane, that in turn, is transduced in ATP synthesis and active transport[3]. With a high pmf of up to 220 mV[29], the NDH-1MS is expected to thermodynamically drive the hydration of CO$_2$ (at 410 ppm; [CO$_2$] = 14 μM; [HCO$_3^-$] = 30 mM[26]; pH = 7), and couple this to pumping of at least three

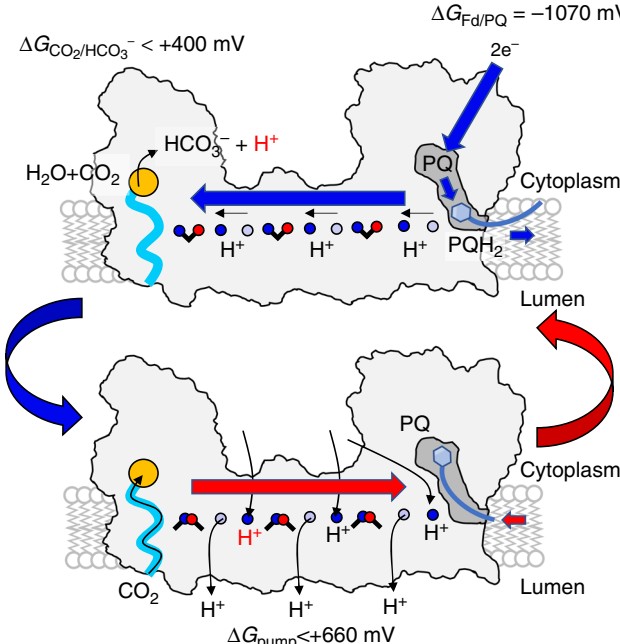

**Fig. 4 Putative mechanism of redox-driven $CO_2$ concentration in the photosynthetic complex I.** Top: PQ reduction to $PQH_2$ leads to dissociation of the quinol to a membrane-bound PQ-binding site[71] that triggers sequential opening of ion-pairs in NdhB, NdhD3, and NdhF3, by coupled hydration, conformational, and protonation changes, protonating terminal proton loading sites (blue forward arrow)[19, 20]. Bottom: Closing of ion-pairs in a sequential electrostatic backward pulse (red backward arrow) releases the protons to the luminal side of the membrane, and allows for re-protonation of the antiporter-like subunits by protons released from the $CO_2$ hydration reaction in the active site of CupA (orange circle). Horizontal proton transfer reactions within each antiporter-like subunit are shown by small horizontal black arrows, and $PQH_2$ (PQ) diffusion out (in) is indicated by small thick blue (red) arrows. $CO_2$ is taken up by the putative gas channel (in light blue) that is expected to be open depending on the ion-pair conformation in NdhF3 (arrow along light blue channel).

protons across the membrane, e.g., by NdhD3, NdhB, and NdhA/C/E/G[11–14,19,20]. However, the free energy available could allow the pump to operate also under considerably lower $CO_2$ concentrations and pH. Under high *pmf* and a reduced quinone pool, the respiratory complex I operates in reverse direction by consuming the proton gradient to oxidize quinol[30], and due to conserved functional elements, a similar operation mode is also expected for NDH-1MS. During such putative backward operation mode, coupled protonation and/or conformational changes at the NdhF3/NdhD3 interface could close the gas channel similar to conformational changes observed in the bacterial complex I[19,31]. Such changes might decouple the pump to avoid the back-reaction of $HCO_3^-$ to $CO_2$, and the diffusion of the latter out of the cell.

We have demonstrated here how unique molecular adaptation enabled the photosynthetic complex I to concentrate $CO_2$ by redox-driven proton-pumping modules and powered by cyclic electron flow around photosystem I. Our findings show how combination of functional modules from different energy transduction machineries[32] allowed primordial organisms to generate functionality and harness energy under changing environmental conditions.

## Methods

**Construction of the *T. elongatus* mutant and purification of the NDH-1MS complex.** The C-terminal tagged CupS-TwinStrep-tag mutant of *T. elongatus* was generated by homologous recombination with plasmid DNA that contains the modified cupS gene and a kanamycin resistance cassette for selection of the mutant allele (see Supplementary Table 4). Cells of the fully segregated mutant were grown in liquid BG-11 medium and expression of the NDH-1MS complex was induced by low $CO_2$ growth conditions. Preparation of the thylakoid membranes and isolation of the tagged complex were performed as described earlier[15,33] with minor modifications. See SI Materials and Methods for details.

**Cryo-EM grid preparation and data collection.** To prepare cryo-EM grids, four microliter of NDH-1MS at 5 mg mL$^{-1}$ were applied to R2/1 grids (Quantifoil) that were glow-discharged for 20 s immediately before use. The sample was incubated 30 s at 100% humidity and 4 °C before blotting for 3.5 s with blotforce 5 and then plunge-frozen into liquid ethane/propane mix cooled by liquid nitrogen using a Vitrobot Mark IV (FEI). Data was acquired on a Titan Krios electron microscope (ThermoFisher, FEI) operated at 300 kV, equipped with a K2 Summit direct electron detector (Gatan) and a GIF quantum energy filter (20 eV) (Gatan). Movies were recorded in counting mode at a pixel size of 1.35 Å per pixel using a cumulative dose of 40.20 e$^-$/Å$^2$ and 50 frames. Data acquisition was performed using SerialEM[34] with four exposures per hole with a target defocus range of 0.5 to 3.0 μm.

**Image processing.** The dose-fractionated movies were gain normalised, aligned and dose-weighted using the motion correction algorithm (Motion Cor2)[35] implemented in RELION 3.0[34,36]. GCTF[37] was used to estimate the defocus values, and particles were selected using Gautomatch. All subsequent processing steps were carried out in RELION 3.0[34]. Following two-dimensional classification, an ab initio model was generated using stochastic gradient descent (SGD). The entire data set (526,925 particles) was subjected to three-dimensional (3D) classification into five classes using the previously determined 60 Å low-pass-filtered ab initio model. A single class consisting of the majority of particles (51.7%) resembled the complex structure. The particles were selected and subsequently sub-classified into 3 classes, using a finer angular sampling (3.75°). The best aligning class consisted of 170,151 particles (62.4%), which were subjected to 3D refinement, yielding an overall resolution of 3.9 Å. The particles were further polished using Bayesian-polishing and the CTF-values were refined on a per particle basis, improving the resolution of the reconstruction to 3.2 Å. The temperature factor (−89.9 Å$^2$) and the resolution of the map were estimated by applying a soft mask around the protein density in the post-processing routine, using the gold standard Fourier shell correlation (FSC) = 0.143 criterion. Directional FSC curves and map anisotropy were assessed using the 3DFSC server[38].

**Model building and validation.** The models for the conserved subunits were taken from the previously published NDH-1L complex (PDB ID: 6HUM)[15]. For the NdhD3 and NdhF3 paralog subunits homology models were generated with the Phyre2 server and fitted as rigid bodies with UCSF Chimera[39]. They were subsequently manually adjusted and rebuilt using Coot[40]. The carbonic anhydrase CupA and its binding partner CupS was built de novo using Coot. The model of NDH-1MS was refined against the cryo-EM map using the phenix.real_space_refine routine in the PHENIX software package[41]. The statistical quality of the final model was assessed using MOLPROBITY[42] and EMRinger[43]. Figures were prepared using PyMOL[44] or UCSF Chimera X[45].

**DFT calculations.** DFT models of the CupA/NdhF3 active site were built based on the cryo-EM atomic coordinates of NDH-1MS. The model comprises the Zn$^{2+}$ ion and its water/OH$^-$ ligands, Phe70, Lys78, Tyr79, His130, Ile131, His133, Arg135, Glu139, and Glu142 of CupA subunit, and Tyr41, Arg37, and Glu114 of the NdhF3 subunit, in addition to two water molecules and the $CO_2$/$HCO_3^-$ substrate. Water molecules between Zn and Tyr41 were modelled based on MD simulations (Supplementary Fig. 5j). The model comprises 185 atoms. DFT models of α-carbonic anhydrase were built based on type II human αCA (PDB ID: 5YUI[46]), comprising the Zn$^{2+}$ ion, and the water/OH$^-$ group, His94 His96, His119, His64, Asn67, Gln92, Glu106, Thr199, Thr200, Val121, Val143, Trp209, and eight water molecules and a $CO_2$ molecule solved in the crystal structure. The model comprises 155 atoms. DFT models of βCA were built based on the PDB ID: 1EKJ[47]. The model comprises the Zn$^{2+}$ ion, Cys160, Cys223, His220, the water/OH$^-$ ligand, Gln151, Asp162, Arg164, Val165, Val184, Asn186, the backbone of Gly224, Tyr205, in addition to four water molecules and the $CO_2$. The model comprises 132 atoms. Protein residues were cut at the Cα-Cβ bond and saturated with hydrogen atoms. The Cβ atoms were kept fixed during geometry optimisations that were performed at the B3LYP-D3/def2-SVP/def2-TZVP(Zn$^{2+}$) level, and single point energies were evaluated at the B3LYP-D3/def2-TZVP level[48–51]. Solvation effects were treated using a polarisable dielectric medium with $\varepsilon = 4$ to model the protein environment[52]. Free energies were computed using the freeh module of TURBOMOLE based on electronic and zero-point energies (ZPE), and enthalpic (ΔH) and entropic (TΔS) effects, estimated at the B3LYP-D3/def2-SVP level by calculating the molecular Hessian. The free energy estimates do not consider dynamical sampling effects. Reaction pathways and transitions states were optimised using a chain-of-states method[53,54]. All QM calculations were performed with TURBOMOLE versions 6.6-7.3[55].

**QM/MM calculations.** Hybrid QM/MM calculations were performed based on our NDH-1MS structure. The models comprise the CupA and NdhF subunits, including

In the figure:
$\Delta G_{Fd/PQ} = -1070$ mV
$2e^-$
$\Delta G_{CO_2/HCO_3^-} < +400$ mV
$HCO_3^- + H^+$
$H_2O + CO_2$
PQ
Cytoplasm
$PQH_2$
Lumen
$H^+$ $H^+$ $H^+$
$CO_2$
$H^+$ $H^+$ $H^+$
$\Delta G_{pump} < +660$ mV

the Zn-binding site, and water/lipid/ions in their 12-Å surroundings. The QM region comprises 135 atoms, including the $Zn^{2+}$-$OH^-$ cofactor, Tyr79, His130, His133, Arg135, Glu139 of CupA subunit, and Arg37 and Tyr41 of the NdhF3 subunit. The QM/MM boundary was described by a link-atoms approach. The model was structure optimised at the QM/MM level using the Adopted Basis Newton-Raphson (ABNR) algorithm, and allowing a 10 Å-sphere within the QM region to relax, followed by QM/MM dynamics at $T = 310$ K using a 1 fs integration timestep. The QM region was described at the B3LYP-D3/def2-SVP(C,H,N,O)/def2-TZVP($Zn^{2+}$) level of theory, and the MM region was described by the CHARMM36 force field[56]. A minimum energy pathway scan between the Zn-$H_2O$ and Tyr41 was performed along a reaction coordinate modelled as linear combination of all proton donor and acceptor distances ($R = r_1 - r_2 + r_3 - r_4$, see Supplementary Fig. 5d). Our reported DFT models predict similar energetics for carbonic anhydrases as those reported in previous studies, cf.[57] and refs. therein and further QM/MM models of these systems were therefore not considered. All QM/MM calculations were performed using an in-house version of the CHARMM/TURBOMOLE interface[58].

**Classical molecular dynamics simulations.** Classical molecular dynamics (MD) simulations were performed based on the cryo-EM structure of NDH-1MS. Missing loops in the antiporter-like and NdhS subunits were modelled with MOD-ELLER.19[59]. The cryo-EM atomic coordinates were relaxed to the electron density map using molecular dynamics flexible fitting (MDFF)[60]. The MDFF relaxed model was further embedded in a POPC membrane and solvated with TIP3P water molecules and 100 mM NaCl concentration. Plastoquinone was modelled based on previous work[16]. The total system comprises ca. 580,000 atoms. MD simulations were performed at constant $T = 310$ K and $p = 1$ bar in an NPT ensemble. The CHARMM36 force field[56] was used to model protein/lipid/water-ion atoms. Parameters for PQ, FeS centres, and the Zn-first coordination sphere were derived from DFT calculations. Long-range electrostatic interactions were described by the particle mesh Ewald (PME) approach. The system was gradually relaxed during the 2.5 ns with harmonic restraints, followed by 20 ns equilibration without restraints. An initial 1 fs integration timestep was used during the equilibration process, and 2 fs timestep during the remaining 250 ns production steps. MD simulations of the $CO_2$ diffusion process were performed based on 20 independent trajectories each 10 ns with the $CO_2$ molecule positioned along the putative cavity in the NdhF3 subunit, that was identified with CAVER[61]. All classical MD simulations were performed with NAMD2[62], and VMD[63] and UCSF Chimera[39] were used for analysis.

**Poisson-Boltzmann continuum electrostatics calculations.** $pK_a$ values of titratable residues in CupA/S/NdhF3 were estimated using the Adaptive Poisson-Boltzmann Solver (APBS)[64] and by Monte Carlo (MC) sampling of $2^N$ protonation states with Karlsberg+[65,66]. PBE/MC calculations can provide accurate estimation of $pK_a$ values in complex biochemical systems[67,68], whereas constant pH-MD simulations that could further enhance the accuracy of the predictions[69,70] are outside the scope of the present work due to the large size and complex surroundings of NDH-1MS. The protein was described using partial charges, embedded in an inhomogeneous dielectric medium with a dielectric constant of 4 inside the protein and 80 for water. The interface between the protein and solvent was calculated by the molecular surface routine, as implemented in APBS, using a solvent probe radius of 1.4 Å, and modelling an implicit ionic strength of 100 mM potassium chloride. The $pK_a$ values were computed as a difference of electrostatic free energy shifts between a model compound in water and the model compound in the protein for 80 structures obtained from the MDFF relaxation.

## Data availability
Data supporting the findings of this manuscript are available from the corresponding authors upon reasonable request. A reporting summary for this Article is available as a Supplementary Information file. The source data underlying Figs. 2b, 3c, 4c–e and Supplementary Figs. 2a, b, 3c, 5c, d are provided as a Source Data file. EMDB accession codes: EMD-10513, PDB ID: 6TJV.

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

## Acknowledgements

We thank H. Wulfhorst for construction of the CupS-TS mutant, M. Völkel for excellent technical assistance and Petra Düchting (Department MGPP, Ruhr University Bochum, Germany) for ICP-OES analysis. J.M.S. and S.K.S. would like to acknowledge E. Conti for unconditional support. J.M.S is grateful to B.D. Engel, J.M. Plitzko, and W. Baumeister for access to the cryo-EM infrastructure and early career support. This work was supported by the European Research Council (ERC) under the European Union's Horizon 2020 research and innovation programme/grant agreement no. 715311 (to V.R.I.K.), the DFG research unit FOR2092 (836/3-2 to M.M.N.), the DFG priority programme 2002 (NO 836/4-1 to M.M.N.) and the DFG grant NO 836/1-1 (to M.M.N.). This work was also supported by Emergence of Life initiative (TRR235 to V.R.I.K.) and by the Knut and Alice Wallenberg Foundation (to V.R.I.K.). Computational resources were provided by SuperMuc/Leibniz-Rechenzentrum (LRZ, grant: pn34he) and PRACE (project nr: 2018194738), awarding us access to MareNostrum at the Barcelona Supercomputing Centre (BSC), Spain (project: pr1ejk).

## Author contributions

J.T. and M.M.N. isolated and characterised the photosynthetic complex I; P.S., A.P.G.H. and V.R.I.K. performed molecular simulations; J.M.S. and S.K.S. prepared cryo-EM grids and collected data; J.M.S. and S.K.S. processed cryo-EM data; J.M.S., S.K.S. and V.R.I.K. built cryo-EM models; J.M.S., S.K.S. and V.R.I.K. analysed and interpreted the cryo-EM models; J.M.S., P.S., J.T., S.K.S., A.P.G.H., G.K., M.M.N. and V.R.I.K analysed the data; J.M.S., M.M.N. and V.R.I.K. directed the project; V.R.I.K. wrote the manuscript with input from all authors.

## Competing interests

The authors declare no competing interests.
