## [Peer Review File · Nature Communications]

Reviewers' Comments:

Reviewer #1:

Remarks to the Author:

The work entitled "Redox-Coupled Proton Pumping Drives Carbon Concentration in the Photosynthetic Complex I" presents a structural study which combines CrioEM data, with Structural modeling, Molecular Dynamics (MD) and QM based calculations to analyze several important mechanistic aspects of structure function relationships in photosynthetic complex I. The results reveal a novel catalytic carbonic anhydrase module (different from alpha and beta carbonic anhydrases), with a unique CupA active site harboring a Zn²⁺ coordinate to His and Arg residues and proposed to be connected through proton exchange/transfer to the proton-pumping modules which are coupled by a 200 Å electrostatic wire to the electron transfer module of the enzyme. A proposed CO₂ channel connecting the protein boundary with the Zn active site is also proposed and several mechanistic conclusions are drawn based on an interpretation of the structure and molecular simulation results.

Although the work is novel, relevant and very interesting, it falls short on many technical aspects (particularly related to the molecular simulations) which prevent publication in its current form. Also, presentation should be improved since many important details have been left out in the main text which are important and necessary to understand and really gauge the impact of the work. Therefore I suggest publication only provided major revisions are performed and simulation aspects improved, Specific points follow:

The paragraph beginning with "To probe the catalytic properties of the CupA module, we performed quantum chemical calculations on the CO₂ hydration reaction and compared this to the reaction energetic in aCA and bCA (Fig. 2b, Extended Data Fig. 5, see Methods)... " needs lots of work. First, in the main text the authors give no detail on the performed calculations, just mentioned that they used QM/MM methods. Authors should properly describe the level of theory, and the type of calculation, particular in relation as to how the energy barriers were obtained. Moreover, some brief explanation of the reaction mechanism for all three enzymes bCA, aCA and CupA is needed in order to properly understand the differences. Key elements are missing, such as who is the key residue acting as base, that removes the proton from the Zn coordinated water that is supposed to initiate the reaction.

Figure 2, needs more explanation. In particular concerning the energy diagram. Authors write DG, however they only computed "energies" with entropic corrections using single points?. Therefore, the profiles are not actual G but E+ZPE. To really compute free energy profiles authors need to perform QM/MM dynamics with some free energy (or PMF) determination method such as Umbrella sampling, Metadynamics or MSMD.

Authors need to provide further detail on all three computed reactions. bCA, aCA and CupA. In supplementary data only the profile for CupA step 1 is shown, no data is provided for bCA and aCA and the remaining steps of CupA. Authors MUST provide free energy (or E+ZPE profiles if they want) of all steps in all three enzymes. Moreover, authors should clearly explain the reaction coordinate used in each reaction step and describe initial conditions.

Also important, since the reaction involves proton transfer. The authors start with Y41 deprotonated, but who takes the proton from Y41?. Who acts as base in a/bCA??. Also related to Y41 and pKa calculations (Figure 5F from extended data). Pka was computed using a continuum model which is not really accurate, authors should actually compute the pKa using better methods such as constant pH Molecular dynamics (See Dr Roibergs works on the subject). Why do they report two pKas???

Concerning Extended data Figure 6. Authors performed MD simulation and looked at the presence of water molecules which they claim show potential proton transfer wires?. However no

quantitative measure of the solvation is given. Authors could compute key position of water molecules inside the protein using IFST method, i.e determining the Water Sites and their properties (see for examples works by Gauto et. al. and/or WATCLUST method doi: 10.1093/bioinformatics/btv411).

Finally, authors claim that protein hosts a tunnel that concentrates CO₂. However, they only provide anecdotal evidence of CO₂ moving inside the proposed tunnel. Authors should compute the free energy profile of CO₂ migration along the tunnel to really measure how much the tunnel contributes (in relative amount) to concentrate CO₂ inside the protein. See for example how this is performed by Myoglobin to concentrate O₂ inside the active site doi: 10.1074/jbc.M112.426056)

Marcelo A. Marti

Reviewer #2:

Remarks to the Author:

Energy-converting NADH:ubiquinone oxidoreductase, respiratory complex I, plays a major role in energy metabolism. In humans, defects in complex I are linked to severe neurodegenerative diseases. During the last years several sub-families of energy converting NADH:ubiquinone oxidoreductase have been identified that evolved from each other and that share a similar structure and a related mechanism. Cyanobacteria and plants contain an energy-converting ferredoxin:plastoquinone oxidoreductase consisting of several subunits with most of them being related to those of mitochondrial complex I. Very recently, the structure of this enzyme complex, also called photosynthetic complex I, from *Thermosynechococcus elongatus* was determined by means of cryo-electron microscopy (cryo-EM).

The manuscript by Schuller et al. describes the cryo-EM structure of the NDH-1MS isoform of the *Thermosynechococcus elongatus* photosynthetic complex I. This isoform is equipped with an additional module catalyzing a carbonic anhydrase (CA) reaction. The authors describe the ferredoxin and the plastoquinone binding sites and several lipids bound to the enzyme complex. Most importantly, they demonstrate that the CA domain is made up of CupA and S on the cytoplasmic side of the membrane. The authors unequivocally show that this domain contains a catalytically active Zn(2+) ion that is ligated in a different way compared to the well-known α and β CAs leading to a different type of mechanism. The proposed mechanism is substantiated by theoretical energy calculations. It is nicely argued that proton translocation by the photosynthetic complex I facilitates the CA reaction by removing the proton, a reaction product. Furthermore, a mechanism is provided that could explain the propagation of the proton translocation along the membrane part of the complex.

The manuscript is very well written, easy to follow and contains novel and highly interesting information. The experiments and the data are sound and well documented. The manuscript benefits very much from the impressive interplay of structural analysis and topical theoretical methods. I just have a few minor points to possibly be considered:

Page 2, third para, first sentence: 'The 0.5 MDa complex has an overall U-shape with 18 isolated subunits.' However, in Extended Data Table 2, 19 subunits are listed.

Page 3, first para, first sentence: 'The CO₂-concentrating CupA/S module (CO₂ uptake/CO₂ hydration protein, ChpY)....' 'up' is underlined but does not contribute to the abbreviation. The abbreviation reads as CO₂ hydration protein, maybe a better wording can be found in order to not confuse the non-experts.

Page 3, second para, last sentence: Is there any evidence that the chlorophyll α/β -carotene play a role in light-induced regulation? If not, I would delete the sentence in the main text and maybe

place it as pure speculation in the legend of the corresponding figure.

Extended Data Fig. 2: I would propose to also label the subunits above NdhF3 in 2a. It should be explained in the legend why NdhN and M appear twice in 2c/e.

Reviewer #3:

Remarks to the Author:

Cyanobacteria contain a set of enzymes related to respiratory complex I, the NDH-1 family, with diverse functions that appear to have evolved to deal with the challenges of oxygenic photosynthesis. A number of papers appeared earlier this year that showed the structure of NDH-1L, which is similar to respiratory complex I, but lacks the first three subunits of this complex that transfer electrons from NADH to the quinone binding site. Instead, a number of specific subunits allow electron transfer from ferredoxin. The paper by Schuller et al. concerns the cryo-EM structure and molecular dynamics studies of NDH-1MS from *Thermosynechococcus elongatus*, an enzyme involved in carbon concentration. The authors show that NDH-1MS contains a noncanonical carbonic anhydrase module, located at the end of the proton transfer chain, and develop a mechanism for its function in concentration CO₂.

The combination of cryo-EM, model building and molecular dynamics studies yields new insights in cyanobacterial photosynthesis that will be of high interest to experts in the field. The paper is overall well-written; it would however benefit from some modifications, in particular concerning the figures and figure legends, as detailed below.

Specific points:

1. p. 3, assignment of the zinc ion in CupA: the cryo-EM density for this ion and its coordination should be shown in a supplementary figure. The zinc ion should also be shown in figure 3a. Further, the text states that the assignment is "supported by multi-element analysis data (Extended Data Fig. 2e)". However, this figure does not show this. There is mention of Zn in Extended Data Fig. 2d, but this is "Inductively-Coupled Plasma Optical Emission Spectrometry". It is totally unclear what this table (2d) shows, and neither this technique nor multi-element analysis are mentioned in the Methods. Further, it is stated that the Zn ligand Arg135 has a pKa <7, but Extended Data fig. 5f states pKa <0.
2. p. 3 third paragraph, the sentence starting "a non-polar tunnel..." is ungrammatical and can't be understood.
3. p. 4 "global dynamics inferred from the cryo-EM map" would be clearer as "global dynamics inferred from the local resolution of the cryo-EM map". The legends of Ext. Data Fig. 7, which show this, are inadequate and should state more explicitly what the color scheme represents.
4. p. 4 "charged residues in the broken helices TM7 and TM12 (Fig. 3)". It is not mentioned which subunit these helices belong to. Figure 3d shows these helices in all three antiporter subunits NdhF3, D3 and B. It would be better to state this in the text and refer to figure 3d instead of figure 3 in full.
5. p. 5 "coupled protonation and/or conformational changes at the NdhF3/NdhD3 interface could close the gas channel and decouple the pump...". This part is very speculative. It is not clear why the reverse reaction would close the channel that is presumably always open in the forward reaction.
6. Figure 1: This figure shows the cryo-EM map of the protein with a simulated, full atomic model of the membrane. Experimentally determined lipids (as seen in Ext. Data Fig. 3) are however not shown. The atomic model distracts from the protein map and partially obscures it (e.g. NdhL). It

would be better to show the EM density map by itself, including observed non-protein density. Further, panel c does not show a back view, but a side view. It may be a good idea to reverse panel a and c, as a has the same orientation as the model in d, which would then be next to it. In panel c, it is not clear what subunit forms the horizontal surface helix (brownish) below NdhK.

7. Figure 3: presumably the CO₂ channel is shown in purple in a, b and c, but this is never stated. It looks very different in b and c, how was the surface determined? Also the scale and orientation of these panels differs considerably and it is unclear how they relate to each other. 3d: what is an "experimentally-refined structure"? There are no densities shown in figure 3, so referring to "further example densities" is strange.

8. Figure 4: This figure could do with better legends to explain the colors and arrows. Why is one of the H⁺ in the lower panel grey?

9. Methods, image processing: "the motion correction algorithm" should be specified as MotionCorr2 and the reference added. FSC does not stand for "Fourier shell correction" but "Fourier shell correlation."

10. Extended Data Fig. 1: twice refinement is misspelled as "refinement".

11. Extended Data Fig. 3: PGT, SQD and DGD should be defined. It is better not to show hydrogens in the atomic models, they clutter the images.

12. Extended Data Fig. 4a: It is unclear what is shown here. Legend for the left panel is missing and the right panel does not show the interaction between the subunits, just the surface.

13. Extended Data Fig. 4b: It is unclear what is shown here. Is orange the pdb ID 2MXA, as stated in the legends, or the cryo-EM structure? The legend also states that the solution structure is cyan, which seems more likely. Further, helix α_3 , mentioned in the legend and in the text on page 3, is not shown. The text mentions movement of α_2 , α_3 , the legend α_1 , α_2 , α_3 .

14. Extended Data Fig. 5f: the legend mention "the latter values". This makes no sense; presumably what is meant is the second figure in each column.

15. Extended Data Fig. 6e: it is not clear what is shown here. What color is mouse and what cyanobacterial?

16. Extended Data Fig. 7: "Dynamics from cryo-EM resolution map" should be "Dynamics from local resolution of the cryo-EM map".

Answer to comments by Reviewer #1

The work entitled “Redox-Coupled Proton Pumping Drives Carbon Concentration in the Photosynthetic Complex I “ presents a structural study which combines CrioEM data, with Structural modeling, Molecular Dynamics (MD) and QM based calculations to analyze several important mechanistic aspects of structure function relationships in photosynthetic complex I. The results reveal a novel catalytic carbonic anhydrase module (different from alpha and beta carbonic anhydrases), with a unique CupA active site harboring a Zn ²⁺ coordinate to His and Arg residues and proposed to be connected through proton exchange/transfer to the proton-pumping modules which are coupled by a 200 Å electrostatic wire to the electron transfer module of the enzyme. A proposed CO₂ channel connecting the protein boundary with the Zn active site is also proposed and several mechanistic conclusions are drawn based on an interpretation of the structure and molecular simulation results.

Although the work is novel, relevant and very interesting, it falls short on many technical aspects (particularly related to the molecular simulations) which prevent publication in its current form. Also, presentation should be improved since many important details have been left out in the main text which are important and necessary to understand and really gauge the impact of the work.

Therefore I suggest publication only provided major revisions are performed and simulation aspects improved, Specific points follow:

Answer: We thank this reviewer for the comments that have helped us to improve our work. Point-to-point answers are given below.

Question: The paragraph beginning with “ To probe the catalytic properties of the CupA module, we performed quantum chemical calculations on the CO₂ hydration reaction and compared this to the reaction energetic in aCA and bCA (Fig. 2b, Extended Data Fig. 5, see Methods)... “ needs lots of work.

First, in the main text the authors give no detail on the performed calculations, just mentioned that they used QM/MM methods.

Answer: Detailed methodology is described in the methods section of the paper that has now been included as part of the main text. Calculations details have also been added to the figure legend and main text:

in the main text: “To probe the catalytic properties of the CupA module, we performed quantum chemical *density functional theory (DFT)* calculations on the CO₂ hydration reaction and compared this to the reaction energetics in α CA and β CA (Fig. 2b, Extended Data Fig. 5, see Methods).”

in Figure 2: “Free energies are reported at the *B3LYP-D3/def2-TZVP/def2-SVP/ ϵ =4 theory level (see Methods).*”

Question: Authors should properly described the level of theory, and the type of calculation, particular in relation as to how the energy barriers were obtained.

Answer: All theory levels are now also described in the main text and clarified in the caption of Figure 2. The methods have also been clarified in the revised method section:

in Figure 2 legend: “Reaction mechanism and free energy profiles for the CO₂ hydration process *based on quantum chemical DFT models*”

“Free energies are reported at the *B3LYP-D3/def2-TZVP/def2-SVP/ε=4 theory level (see Methods).*”

in methods: “Reaction pathways *and transitions states* were optimized using a chain-of-states method related to the zero-temperature string method^{54,55}”

Question: Moreover, some brief explanation of the reaction mechanism for ll three enzymes βCA, αCA and CupA is needed in order to proper understand the differences. Key elements are missing, such as who is the key residue acting as base, that removes the proton from the Zn coordinated water that is supposed to initiate the reaction.

Answer: We have now clarified key elements of the three enzymes in the main text: “In CupA, *proton transfer from the Zn-bound water to Tyr41 is slightly exergonic in our DFT models, whereas the nucleophilic attack of the Zn-bound OH to the CO₂ has a barrier of ca. 15 kcal mol⁻¹, predicting that catalysis takes place in the millisecond timescale. The involvement of Tyr41, similarly to Tyr205 in βCA,²⁴ is also supported by QM/MM models (Extended Data Figure 5e) as well as by MD simulations and pK_a calculations (Extended Data Figure 5f).*”

For the αCA and βCA, the base is described in the main text: “In αCA (βCA), the rate-limiting reaction barrier of 11 (12) kcal mol⁻¹ is connected with pT to His64 (Tyr205), which compares well with the experimentally-observed barriers of ca. 10-12 kcal mol⁻¹ (Fig. 2b).⁸”

Question: Figure 2, needs more explanation. In particular concerning the energy diagram. Authors write DG, however they only computed “energies” with entropic corrections using single points?. Therefore, the profiles are not actual G but E+ZPE.

Answer: Figure 2 shows free energy profiles computed from quantum chemical DFT models, as now clarified in the figure legend. As explained in the methods section, the free energies are computed as $\Delta H - T\Delta S + \Delta ZPE$ based on the electronic energy and the molecular Hessian. We have clarified in the methods sections that the free energies do not account for dynamical sampling of environmental effects. We now clarify in the methods section:

“Free energies were computed using the *freeh* module of TURBOMOLE based on the electronic and zero-point energies (ZPE), and enthalpic (ΔH) and entropic ($T\Delta S$) effects, estimated at the

B3LYP-D3/def2-SVP level by calculating the molecular Hessian. The free energy estimates do not consider dynamical sampling effects.”

Question: To really compute free energy profiles authors need to perform QM/MM dynamics with some free energy (or PMF) determination method such as Umbrella sampling, Metadynamics or MSMD.

Answer: Free energies can also be estimated quantum chemically. We have clarified in the methods section that there is no dynamical sampling. Detailed QM/MM free energy exploration of the complete NDH-1MS/CupA reaction mechanism will be explored in future work, when the system can be explored at the same time with biochemical and biophysical experiments to assess the predicted mechanisms.

Question: Authors need to provide further detail on all three computed reactions. bCA, aCA and CupA. In supplementary data only the profile for CupA step 1 is shown, no data is provided for bCA and aCA and the remaining steps of CupA. Authors MUST provide free energy (or E+ZPE profiles if they want) of all steps in all three enzymes. Moreover, authors should clearly explain the reaction coordinate used in each reaction step and describe initial conditions.

Answer: We have modified Figure 2 to better emphasize that our study focuses on the novel CupA module, rather than on carbonic anhydrases. However, for comparison, we show free energy profiles for all three enzymes in Figure 2, as now clarified in the revised legend. QM/MM free energies for all three enzymes is outside the scope of the present work, particularly since the canonical carbonic anhydrase mechanisms is rather well-understood. The QM/MM energy profiles for CupA were meant to further explore the initial deprotonation reaction. We have added a citation to previous QM/MM work on carbonic anhydrases in the methods section: *“Our reported DFT models predict similar energetics for carbonic anhydrases as those reported in previous studies, ^{cf. 58 and refs therein} and further QM/MM models of these systems were therefore not considered.”*

Question: Also important, since the reaction involves proton transfer. The authors start with Y41 deprotonated, but who takes the proton from Y41?. Who acts as base in a/bCA??. Also related to Y41 and pKa calculations (Figure 5F from extended data). Pka was computed using a continuum model which is not really accurate, authors should actually compute the pKa using better methods such as constant pH Molecular dynamics (See Dr Roibergs works on the subject). Why do they report two pKas???

Answer: Continuum electrostatic calculations with Monte Carlo sampling can provide accurate ways to assess pK_a in large protein models, for which we have provided citations. Our pK_a calculations suggest that Y41 could be deprotonated at physiological pH. We now also provide further MD simulations in both the protonated and deprotonated forms of Y41, suggesting that the latter form might better match the distances observed in our cryo-EM structure. This additional MD simulation is now included in the Supplementary material (Figure S5f).

As stated in the legend of Figure S5, the two states correspond to microstates obtained with Arg135 fixed in its deprotonated states or fully titrating the residue. Both simulations suggest that Arg135 and Tyr41 could be deprotonated.

Although we agree that constant pH-MD simulations provide important developments in the field of computational biochemistry, such simulations are currently outside the scope of the present work. Many constant pH-MD simulations are carried out using implicit solvent models that is not expected to accurately capture the NDH1-MS dynamics at its challenging protein-water-lipid interface. Another drawback is that our system comprises more than half a million atoms and around 1000 titratable groups, which is not expected to yield converged sampling on accessible timescales.

At present, our combined pK_a calculations, QM models, and MD simulations support that Y41 could be involved in the proton transfer reactions, similarly as in β CA. We have also clarified in the revised text that the proton acceptors are His64 and Tyr205 in α CA and β CA, respectively.

Revisions in main text: *“The involvement of Tyr41, similarly to Tyr205 in β CA,²⁴ is also supported by QM/MM models (Extended Data Figure 5e) as well as by MD simulations and pK_a calculations (Extended Data Figure 5f).”*

24. Rowlett, R. S. *et al.* Kinetic characterization of wild-type and proton transfer-impaired variants of β -carbonic anhydrase from *Arabidopsis thaliana*. *Arch. Biochem. Biophys.* **404**, 197–209 (2002).

Revisions in the methods section: *“PBE/MC calculations can provide accurate estimation of pK_a values in complex biochemical systems,^{68,69} whereas constant pH-MD simulations that could further enhance the accuracy of the predictions^{70,71} are outside the scope of the present work due to the large size and complex surroundings of NDH-IMS.”*

68. Meyer, T. & Knapp, E. W. pK_a Values in Proteins Determined by Electrostatics Applied to Molecular Dynamics Trajectories. *J. Chem. Theory Comput.* **11**, 2827–2840 (2015).

69. Kieseritzky, G. & Knapp, E. W. Improved pK_a prediction: Combining empirical and semimicroscopic methods. *J. Comput. Chem.* **29**, 2575–2581 (2008).

70. Baptista, A. M., Teixeira, V. H. & Soares, C. M. J. Constant-pH molecular dynamics using stochastic titration. *Chem. Phys.* **117**, 4184–4200 (2002).

71. Swails, J.M. York D. M. & Roitberg A.E. Constant pH Replica Exchange Molecular Dynamics in Explicit Solvent Using Discrete Protonation States: Implementation, Testing, and Validation. *J. Chem. Theory Comput.* **10**, 1341–1352 (2014).

Question: Concerning Extended data Figure 6. Authors performed MD simulation and looked at the presence of water molecules which they claim show potential proton transfer wires?. However no quantitative measure of the solvation is given. Authors could compute key position of water molecules inside the protein using IFST method, i.e determining the Water Sites and their properties (see for examples works by Gauto *et. al.* and/or WATCLUST method doi: 10.1093/bioinformatics/btv411).

Answer: We thank this reviewer for suggesting the IFST or WATCLUST programs. We now report the number of water molecules observed during the MD simulations in the proton channels based on our in-house analysis scripts (Extended Figure 6f). This analysis shows that the hydration increases during the MD simulations, consistent with our previous observation on the bacterial complex I (PNAS 2014, 2017; BBA 2018, 2019).

Question: Finally, authors claim that protein hosts a tunnel that concentrates CO₂. However, they only provide anecdotal evidence of CO₂ moving inside the proposed tunnel. Authors should compute the free energy profile of CO₂ migration along the tunnel to really measure how much the tunnel contributes (in relative amount) to concentrate CO₂ inside the protein. See for example how this is performed by Myoglobin to concentrate O₂ inside the active site doi: 10.1074/jbc.M112.426056)

Marcelo A. Marti

Answer: We respectfully disagree with the reviewer that the evidence is anecdotal, as we observe a complete CO₂ conduction during our unbiased MD simulations across the membrane domain to the Zn-site.

Obtaining accurate free energies is however highly challenging for such 50 Å long reaction pathway in a 0.5 million atom system, and task that we estimate would alone require at least 5 Mio CPUh to achieve. This is therefore outside the scope of the present work that focuses on general structure and function of the CO₂-concentrating photosynthetic complex I. We appreciate the reviewer's suggestion to employ non-equilibrium pulling simulations and the Jarzynski equality. However, experience in the field suggest that it is highly challenging to obtain converged free energies for complex biological systems using such approaches (*cf.* work by Dellago and Hummer).

Following the reviewer's suggestion, we have extended the CO₂ simulations to 200 ns and now show the probability distribution and the resulting free energy profile obtained from the unbiased MD simulations in the revised Figure 3. However, we also emphasize in the figure legend that sampling gaps due at regions with putative gating function, introduce a large error source in the derived *pmf*.

Answer to comments by Reviewer #2

Energy-converting NADH:ubiquinone oxidoreductase, respiratory complex I, plays a major role in energy metabolism. In humans, defects in complex I are linked to severe neurodegenerative diseases. During the last years several sub-families of energy converting NADH:ubiquinone oxidoreductase have been identified that evolved from each other and that share a similar structure and a related mechanism. Cyanobacteria and plants contain an energy-converting ferredoxin:plastoquinone oxidoreductase consisting of several subunits with most of them being related to those of mitochondrial complex I. Very recently, the structure of this enzyme complex, also called photosynthetic complex I, from *Thermosynechococcus elongatus* was determined by means of cryo-electron microscopy (cryo-EM).

The manuscript by Schuller et al. describes the cryo-EM structure of the NDH-1MS isoform of the *Thermosynechococcus elongatus* photosynthetic complex I. This isoform is equipped with an additional module catalyzing a carbonic anhydrase (CA) reaction. The authors describe the ferredoxin and the plastoquinone binding sites and several lipids bound to the enzyme complex. Most importantly, they demonstrate that the CA domain is made up of CupA and S on the cytoplasmic side of the membrane. The authors unequivocally show that this domain contains a catalytically active Zn(2+) ion that is ligated in a different way compared to the well-known α and β CAs leading to a different type of mechanism. The proposed mechanism is substantiated by theoretical energy calculations. It is nicely argued that proton translocation by the photosynthetic complex I facilitates the CA reaction by removing the proton, a reaction product. Furthermore, a mechanism is provided that could explain the propagation of the proton translocation along the membrane part of the complex.

The manuscript is very well written, easy to follow and contains novel and highly interesting information. The experiments and the data are sound and well documented. The manuscript benefits very much from the impressive interplay of structural analysis and topical theoretical methods. I just have a few minor points to possibly be considered:

Answer: We thank the reviewer for the positive comments that helped us to further improve our work.

Question: Page 2, third para, first sentence: ‘The 0.5 MDa complex has an overall U-shape with 18 isolated subunits.’ However, in Extended Data Table 2, 19 subunits are listed.

Answer: Extended data Table 2 lists 19 subunits as it also shows the data for subunit NdhV. We have clarified in the main text that the structure of this subunit could not be refined in the cryo-EM structure. We now write: “The 0.5 MDa complex has an overall U-shape with 19 isolated subunits (Extended Data Fig. 2 and Extended Data Table 2). *The structure of NdhV could not be resolved.*”

Question: Page 3, first para, first sentence: ‘The CO₂-concentrating CupA/S module (CO₂ uptake/CO₂ hydration protein, ChpY)...’ ‘up’ is underlined but does not contribute to the abbreviation. The abbreviation reads as CO₂ hydration protein, maybe a better wording can be found in order to not confuse the non-experts.

Answer: “*up*” contributes to the “Cup” abbreviation that derives from CO₂ uptake protein that we have now clarified in the main text. Both Cup/Chp abbreviations are used in the literature, and we therefore wish to keep both names:

“The CO₂-concentrating CupA/S module (CO₂ uptake, Cup/CO₂ hydration protein, ChpY)”

Question: Page 3, second para, last sentence: Is there any evidence that the chlorophyll α/β -carotene play a role in light-induced regulation? If not, I would delete the sentence in the main text and maybe place it as pure speculation in the legend of the corresponding figure.

Answer: We do not have direct evidence of the participation of the chlorophyll α/β -carotene in light induced regulation. However, as the site is clearly visible in our structure, we have kept the sentence in the main text, but emphasized the lacking experimental evidence:

*“Despite lacking experimental evidence of its functionality, we speculate that the motif could be involved in light-triggered regulation of the CA activity and/or to provide structural stability. A similar Chl α/β -carotene motif with unknown function is also found in cytochrome *b₆f*.²³”*

23. Kurisu, G., Zhang, H., Smith, J. L. & Cramer, W. A. Structure of the Cytochrome *b₆f* Complex of Oxygenic Photosynthesis: Tuning the Cavity. *Science*. **302**, 1009–1014 (2003).

Question: Extended Data Fig. 2: I would propose to also label the subunits above NdhF3 in 2a. It should be explained in the legend why NdhN and M appear twice in 2c/e.

Answer: A clear assignment of the bands in the NdhF region is difficult in our experience, as we often observe various aggregates of membrane intrinsic proteins in the upper part of the gel that are not dissolved during SDS-PAGE analysis. Most probably the additional bands above NdhF3 belong to aggregates and do not reflect individual (novel) subunits. This would be unlikely, as all subunits that are visible in the structure, have also been identified by MS analysis (see Table 2). NdhN and other subunits appear twice in 2c/e due to multiple charge states of the corresponding ions. The figure legend was changed accordingly.

Answer to comments by Reviewer #3

Cyanobacteria contain a set of enzymes related to respiratory complex I, the NDH-1 family, with diverse functions that appear to have evolved to deal with the challenges of oxygenic photosynthesis. A number of papers appeared earlier this year that showed the structure of NDH-1L, which is similar to respiratory complex I, but lacks the first three subunits of this complex that transfer electrons from NADH to the quinone binding site. Instead, a number of specific subunits allow electron transfer from ferredoxin. The paper by Schuller et al. concerns the cryo-EM structure and molecular dynamics studies of NDH-1MS from *Thermosynechococcus elongatus*, an enzyme involved in carbon concentration. The authors show that NDH-1MS contains a noncanonical carbonic anhydrase module, located at the end of the proton transfer chain, and develop a mechanism for its function in concentration CO₂.

The combination of cryo-EM, model building and molecular dynamics studies yields new insights in cyanobacterial photosynthesis that will be of high interest to experts in the field. The paper is overall well-written; it would however benefit from some modifications, in particular concerning the figures and figure legends, as detailed below.

We thank the reviewer for the excellent comments that have helped us to improve our work.

Question 1.

Specific points:

p. 3, assignment of the zinc ion in CupA: the cryo-EM density for this ion and its coordination should be shown in a supplementary figure. The zinc ion should also be shown in figure 3a.

Answer: The Zn²⁺ ion and its cryoEM density has now been included in the revised Figure 2a. The Zn ion together with the cryoEM map is also shown more clearly in Figure 3a.

Supplementary figure

Question: Further, the text states that the assignment is “supported by multi-element analysis data (Extended Data Fig. 2e)”. However, this figure does not show this. There is mention of Zn in Extended Data Fig. 2d, but this is “Inductively-Coupled Plasma Optical Emission Spectrometry”. It is totally unclear what this table (2d) shows, and neither this technique nor multi-element analysis are mentioned in the Methods.

Answer: We have used Inductively-Coupled Plasma Optical Emission Spectrometry to quantify zinc in the NDH-1MS sample. A method description was added to SI-methods section and the results are now briefly explained in the legend of Extended Data Fig. 2:

“Zinc quantification by Inductively-Coupled Plasma Optical Emission Spectrometry (ICP-OES) according to Ref.⁶⁶ “Zinc was detected at three specific wavelengths (2025 nm, 2062 nm and 2138 nm) and quantified in control (buffer only) and NDH-1MS sample after calibration with a

zinc standard. Comparison with the protein concentration indicates a protein-to-zinc ratio of ~0.7.”

Question: Further, it is stated that the Zn ligand Arg135 has a pKa <7, but Extended Data fig. 5f states pKa <0.

Answer: predicted pK_a values that are in the extreme range often indicate that the residues strongly favors the deprotonated/protonated state. We have reformulated the sentence: *”and our electrostatic calculations suggest that the residue is neutral with pK_a << 7”*

Question 2. p. 3 third paragraph, the sentence starting “a non-polar tunnel...” is ungrammatical and can’t be understood.

Answer: We have revised the sentence accordingly: *“A non-polar tunnel starting from NdhF3 that leads to the Zn²⁺-site ...”*

Question 3. p. 4 “global dynamics inferred from the cryo-EM map” would be clearer as “global dynamics inferred from the local resolution of the cryo-EM map”. The legends of Ext. Data Fig. 7, which show this, are inadequate and should state more explicitly what the color scheme represents.

Answer: the sentence has been revised: *“with global dynamics inferred from the local resolution of the cryo-EM map,”*

We have now also clarified the color scheme in the revised Ext. Data Fig. 7 figure and revised the legend accordingly: *“Extended Data Fig. 7 | Dynamics of NDH-1MS inferred from the local resolution of the cryo-EM map and from MD simulations. a) The resolution was estimated using the local resolution function in RELION with default parameters and plotted using UCSF Chimera. Units are in (Å). b) Root-mean-square-fluctuations (RMSF, in Å) obtained from 250 ns MD simulations of NDH-1MS.”*

Question 4. p. 4 “charged residues in the broken helices TM7 and TM12 (Fig. 3)”. It is not mentioned which subunit these helices belong to. Figure 3d shows these helices in all three antiporter subunits NdhF3, D3 and B. It would be better to state this in the text and refer to figure 3d instead of figure 3 in full.

Answer: To clarify the subunits, we have now revised the sentence: *“The proton channels are established across the membrane around charged residues in the broken helices TM7 and TM12 of the antiporter-like subunits NdhB, NdhD3,^{19,20} and also in NdhA/C/E/G (Fig 1d, Fig. 3, Extended Data Fig. 6a-d).”*

Question 5. p. 5 “coupled protonation and/or conformational changes at the NdhF3/NdhD3 interface could close the gas channel and decouple the pump...”. This part is very speculative. It is not clear why the reverse reaction would close the channel that is presumably always open in the forward reaction.

Answer: We agree with the reviewer that the model is speculative. We have clarified the background better in the revised text:

”During such putative backward operation mode, coupled protonation and/or conformational changes at the NdhF3/NdhD3 interface could close the gas channel similar to conformational changes observed in the bacterial complex I.^{19,31} Such changes might decouple the pump to avoid the back-reaction of HCO₃⁻ to CO₂, and the diffusion of the latter out of the cell.”

19. Kaila, V. R. I. Long-range proton-coupled electron transfer in biological energy conversion: Towards mechanistic understanding of respiratory complex I. *J. R. Soc. Interface* **15**, 20170916 (2018).

31. Di Luca, A., Mühlbauer, M. E., Saura, P. & Kaila, V. R. I. How inter-subunit contacts in the membrane domain of complex I affect proton transfer energetics. *BBA - Bioenerg.* **1859**, 734–741 (2018).

Question 6. Figure 1: This figure shows the cryo-EM map of the protein with a simulated, full atomic model of the membrane. Experimentally determined lipids (as seen in Ext. Data Fig. 3) are however not shown. The atomic model distracts from the protein map and partially obscures it (e.g. NdhL). It would be better to show the EM density map by itself, including observed non-protein density. Further, panel c does not show a back view, but a side view. It may be a good idea to reverse panel a and c, as a has the same orientation as the model in d, which would then be next to it. In panel c, it is not clear what subunit forms the horizontal surface helix (brownish) below NdhK.

Answer: We have changed Figure 1 according to the reviewer’s suggestion. To this end, we removed the modelled lipids from the protein density map, and highlighted the experimentally resolved lipids and cofactors around the protein.

Question 7. Figure 3: presumably the CO₂ channel is shown in purple in a, b and c, but this is never stated. It looks very different in b and c, how was the surface determined? Also the scale and orientation of these panels differs considerably and it is unclear how they relate to each other. 3d: what is an “experimentally-refined structure”? There are no densities shown in figure 3, so referring to “further example densities” is strange.

Answer: The CO₂ tunnel shown in a and b was determined using CAVER that we have clarified in the revised figure legend. The surface shown in panel c corresponds to the average of CO₂ molecules positions sampled during an MD simulation, and its orientation relative to the other panels has been clarified in the revised figure. Panel 3d, showing the “experimental structure” of the chlorophyll a/β-carotene cofactors, has been moved to Figure 1.

Question 8. Figure 4: This figure could do with better legends to explain the colors and arrows. Why is one of the H⁺ in the lower panel grey?

Answer: The pumped protons are all colored in the same way, and the arrows have been clarified in the revised legend:

“...from the CO₂ hydration reaction in the active site of CupA (orange circle). Horizontal proton transfer reactions within each antiporter-like subunit are shown by small horizontal black arrows, and PQH₂ (PQ) diffusion out (in) is indicated by small thick blue (red) arrows. CO₂ is taken up by the putative gas channel (in light blue) that is expected to be open depending on the ion-pair conformation in NdhF3 (arrow along light blue channel).”

Question 9. Methods, image processing: “the motion correction algorithm” should be specified as MotionCorr2 and the reference added.

Answer: MotionCorr2 and the reference has now been added.

Question FSC does not stand for “Fourier shell correction” but “Fourier shell correlation.”

Answer: This has now been corrected in the revised text.

Question 10. Extended Data Fig. 1: twice refinement is misspelled as “refinement”.

Answer: We have corrected the misspelled words in the revised text.

Question 11. Extended Data Fig. 3: PGT, SQD and DGD should be defined. It is better not to show hydrogens in the atomic models, they clutter the images.

Answer: We have removed the hydrogens from the figure and clarified the abbreviations in the revised legend: “SQD – sulfoquinovosyl diglyceride; DGD – digalactosyl diacylglycerol; PGT – phosphatidylglycerol.”

Question 12. Extended Data Fig. 4a: It is unclear what is shown here. Legend for the left panel is missing and the right panel does not show the interaction between the subunits, just the surface.

Answer: The figure shows the electrostatic potential at the protein surface, with a detail of the CupA/NdhF3 surfaces and contact. The surface is colored from negative (red) to positive (blue) potential. This is now better explained in the revised figure caption: “CupA structure and interactions. a) Electrostatic potential (in kcal/mol e) at the surface of the complete NDH-1MS (left) and closeup of the CupA/NdhF3 interface (right). The negative area at the bottom of CupA (red) electrostatically interacts with the positive area (blue) at the top of NdhF3 (CupA-NdhF3 contact shown by a thick black line).”

Question 13. Extended Data Fig. 4b: It is unclear what is shown here. Is orange the pdb ID 2MXA, as stated in the legends, or the cryo-EM structure? The legend also states that the solution structure is cyan, which seems more likely. Further, helix α_3 , mentioned in the legend and in the text on page 3, is not shown. The text mentions movement of α_2 , α_3 , the legend α_1 , α_2 , α_3 .

Answer: We have clarified in the revised figure that it shows an overlap of the current cryo-EM structure (in orange) and a previous solution structure (in cyan). We have also highlighted the position of helix α_3 and revised the figure legend: “**b** *CupS* undergoes conformational changes from its solution structure (*CupS* in cyan, PDB ID: 2MXA²¹) upon binding to *CupA* (*CupS* in orange). The figure shows how helices α_1 , α_2 , and α_3 in the N-terminal region, and α_5 and α_6 in the middle region, close upon the β -sheet.”

Question 14. Extended Data Fig. 5f: the legend mention “the latter values”. This makes no sense; presumably what is meant is the second figure in each column.

Answer: To clarify this, we have revised the legend: “The *second set of* values correspond to calculations where R135 has been fixed in its deprotonated state (indicated with asterisk).”

Question 15. Extended Data Fig. 6e: it is not clear what is shown here. What color is mouse and what cyanobacterial?

Answer: We have now clarified in the revised figure legend that the mouse structure is shown to the right and the cyanobacterial structure to the left. For the mouse structure we further show the deactive state in grey and the active state in purple.

“**e** TM3 of NdhG in NDH-IMS (grey, left). TM3 helix of ND6 of complex I from *Mus musculus* (right) in the deactive (grey) and active (purple) states”

Question 16. Extended Data Fig. 7: “Dynamics from cryo-EM resolution map” should be “Dynamics from local resolution of the cryo-EM map”.

Answer: The sentence has been corrected in the revised text: “Dynamics of NDH-IMS inferred from the local resolution of the cryo-EM map and from MD simulations.”

Reviewers' Comments:

Reviewer #1:

Remarks to the Author:

The revised version of the work "Redox-Coupled Proton Pumping Drives Carbon Concentration in the Photosynthetic Complex I" presents a significant improved and corrected version of the manuscript. Authors have addressed and/or addressed most of my previous concerns as well as those presented by the other reviewers.

Specifically, they have:

- i) added, as requested, a detailed methodology in the methods section now included as part of the main text, which properly describes theory levels and methodology
- ii) Clarified key elements of the three enzymes that are compared in the work (α CA, β CA and current enzyme).
- iii) Added proper description that energy calculations correspond to $\Delta H - T\Delta S + \Delta ZPE$ based on the electronic energy and the molecular Hessian.
- iv) Better explained the role of the titrable active site residues and pKa calculation
- v) Increased sampling and analysis of the waters present in the proton channels
- vi) Provided estimate of the Free Energy Profile of CO₂ migration along the proposed channel.

Therefore, I believe the present work is suitable for publication. I only have a minor comment (described below) related to Figure 2b,.

In the text describing Figure 2 authors state (starting line 123) "The CO₂ hydration is initiated by proton transfer (pT) from the Zn-bound water molecule followed by a nucleophilic attack.. of the hydroxide on CO₂. In α CA (β CA), the rate limiting reaction barrier of 11 (12) kcal mol⁻¹ is connected with pT to His64 (Tyr205), ... In CupA, proton transfer from Zn-bound water to Tyr41 is slightly exergonic" Therefore, if I understood it correctly, in CupA Zn bound water (which then acts as the nucleophile) transfer the proton to Tyr41 (in α CA / β CA the acceptor being His64/Tyr205). However, in Figure 2B, the Zn bound water transfer the proton to ANOTHER water molecule, which then transfers it to Tyr/His/Tyr. So, are there 1 or 2 water molecules? In line with this point in Figure 2A waters are missing. I strongly believe they should be shown and the issue needs to be clarified.

Reviewer #2:

Remarks to the Author:

I think that the authors have addressed the questions raised by all reviewers adequately, so that the revised version of the manuscript can be accepted.

Reviewer #3:

Remarks to the Author:

The authors have answered all questions to my satisfaction and especially the figures have been improved considerably. I support publication of the paper.

".

Reply to reviewer's point:

Question: In the text describing Figure 2 authors state (starting line 123) “ The CO₂ hydration is initiated by proton transfer (pT) from the Zn-bound water molecule followed by a nucleophilic attack.. of the hydroxide on CO₂ . In α CA (β CA), the rate limiting reaction barrier of 11 (12) kcal mol⁻¹ is connected with pT to His64 (Tyr205), ... In CupA, proton transfer from Zn-bound water to Tyr41 is slightly exergonic” Therefore, if I understood it correctly, in CupA Zn bound water (which then acts as the nucleophile) transfer the proton to Tyr41 (in α CA / β CA the acceptor being His64/Tyr205).

However, in Figure 2B, the Zn bound water transfer the proton to ANOTHER water molecule, which then transfers it to Tyr/His/Tyr. So, are there 1 or 2 water molecules?. In line with this point in Figure 2A waters are missing. I strongly believe they should be shown and the issue needs to be clarified.

Answer: We thank the reviewer for the additional comment. The proton transfer from the Zn-bound water molecule to Tyr41 takes place by two water molecule that are formed during our MD simulations. We have now clarified this in the Methods section, and show the water count between the Zn/Tyr41 based on MD simulations in Supplementary Figure 5j. Figure 2A shows the experimentally refined structure, but the resolution is unfortunately not high enough to resolve water molecules. The rather strong Zn-bound density has been modeled as a water ligand, although the character is not fully clear based on the maps. This has now also been clarified in the Figure legend.